# Fructose-Rich Diet Is a Risk Factor for Metabolic Syndrome, Proximal Tubule Injury and Urolithiasis in Rats

**DOI:** 10.3390/ijms23010203

**Published:** 2021-12-24

**Authors:** Mariusz Flisiński, Andrzej Brymora, Natalia Skoczylas-Makowska, Anna Stefańska, Jacek Manitius

**Affiliations:** 1Department of Nephrology, Hypertension and Internal Medicine, Collegium Medicum in Bydgoszcz, Nicolaus Copernicus University in Toruń, 85-094 Bydgoszcz, Poland; andrzej.brymora@cm.umk.pl (A.B.); synowie2@o2.pl (J.M.); 2Department of Clinical Pathology, Collegium Medicum in Bydgoszcz, Nicolaus Copernicus University in Toruń, 85-094 Bydgoszcz, Poland; n.makowska@cm.umk.pl; 3Department of Laboratory Medicine Collegium Medicum in Bydgoszcz, Nicolaus Copernicus University in Toruń, 85-094 Bydgoszcz, Poland; zuzanna@cm.umk.pl

**Keywords:** fructose-rich diet, metabolic syndrome, proximal tubule injury, urolithiasis, rats

## Abstract

Excessive consumption of fructose (FR) leads to obesity, metabolic syndrome (MS) and insulin resistance, which are known risk factors for kidney stones. The epidemiological study has suggested the association between fructose consumption and urolithiasis, but the precise mechanism is still not well understood. Male Wistar rats were assigned for 8 weeks to three groups with different FR content in diet: RD (*n* = 5)—regular diet with a FR <3%; F10 (*n* = 6)—regular diet with an addition of 10% Fr in drinking water; F60 (*n* = 5)—60% FR as a solid food. Serum concentration of FR, creatinine (Cr), insulin (Ins), triglycerides (Tg), homocysteine (HCS), uric acid (UA), calcium (Ca), phosphate (Pi), magnesium (Mg) and sodium (Na) were measured. Based on 24 h urine collection the following tests were performed: urine pH, proteinuria (PCR), excretion of N-Acetyl-(D)-Glucosaminidase (NAG), monocyte chemoattractant protein (MCP-1), uric acid (uUAEx), phosphate (uPiEx), calcium (uCaEx), magnesium (uMgEx) and sodium (uNaEx). The creatinine clearance (CrCl) was calculated. Calcium deposits in kidney sections were examined using hematoxylin and eosin (HE) and von Kossa stains. The rats on F10 and F60, as compared to the RD diet, showed a tendency for lower CrCl, higher HCS level and some features of MS as higher Ins and TG levels. Interestingly, F10 (fluid) versus F60 (solid) diet led to higher serum Ins levels. F10 and F60 versus RD demonstrated higher urinary excretion of MCP-1 and NAG which were suggestive for inflammatory injury of the proximal tubule. F10 and F60 as compared to RD showed significantly lower uUAEx, although there were no differences in clearance and fractional excretion of UA. F60 versus RD induced severe phosphaturia (>30×) and natriuria (4×) and mild calciuria. F10 versus RD induced calciuria (3×), phosphaturia (2×) and mild natriuria. Calcium phosphate stones within the tubules and interstitium were found only in rats on FR diet, respectively, in two rats from the F10 group and another two in the F60 group. The rats which developed stones were characterized by significantly higher serum insulin concentration and urinary excretion of calcium and magnesium. A fructose-rich diet may promote development of calcium stones due to proximal tubule injury and metabolic syndrome.

## 1. Introduction

Fructose (FR) is a monosaccharide occurring naturally in fruits and honey. Its consumption has been increasing dramatically worldwide primarily due to the addition of sucrose (a disaccharide containing 50% fructose and 50% glucose) and high-fructose corn syrup (HFCS) to foods and soft drinks. It is estimated that the FR consumption has increased by 2000% in the US since HFCS introduction in 1967 [1]. Today, the mean FR intake is about 74 g/day, which is twenty-five times greater than the average intake in the United Kingdom in 1700s [2]. A similar trend has been observed in European countries where total sugar consumption contributes to 15 to 25% of energy supply, with relatively higher intake of added sugars in children compared to adults [3]. In our previous study on a group of patients with stage 2 and 3 chronic kidney disease (CKD), the daily FR intake was 59 ± 22 g, where added sugars constituted 63% [4]. 

Metabolism of FR is distinct from other sugars. It quickly undergoes phosphorylation to fructose-1-phosphate by fructokinase in the liver due to a lack of negative feedback. As a results, the intracellular phosphate (P) and adenosine triphosphate (ATP) depletion can occur transiently with further generation of adenosine monophosphate (AMP), which is then metabolized to inosine monophosphate and eventually to uric acid (UA) [5]. The transient ATP depletion shows some similarities to ischemia and can lead to termination of protein synthesis, oxidative stress and inflammation [6,7]. UA causes reduction in endothelial nitric oxide (NO) level, the key mediator of insulin action, which in turn disturbs blood flow and glucose uptake in skeletal muscles leading to insulin resistance and development of metabolic syndrome (MS) [8]. 

The clinical and epidemiological evidence also suggests an association between FR consumption and obesity [1], pancreatic beta cell dysfunction [9], type 2 diabetes [1,10], gout, hypertension, chronic kidney disease and cardiovascular disease [11,12]. In developed countries, along with the epidemic of obesity and diabetes, there is also a growing incidence of kidney stone disease in both adults [13] and children [14]. The study by Taylor and Curhan has shown the association between FR consumption from added sugars and kidney stones in three large cohorts of patients. The relative risk of nephrolithiasis was significantly higher in participants with the highest intake of total and free FR compared to those with the lowest intake based on semi-quantitative food frequency questionnaires [15]. It is hypothesized that this association may be caused by increased urinary excretion of calcium (uCaEx) [16], uric acid [17] and oxalate [18], or by the induction of insulin resistance, which is associated with low urine pH and precipitation of urate salts [19]. A low content of magnesium (Mg) and citrates, which act as crystallization inhibitors, as well as low fibre content in the Western diet, together with processed foods can also promote stone formation in the kidneys and urinary tract [16,20]. The prevalence of nephrolithiasis is higher in patients with obesity, metabolic syndrome and type 2 diabetes. At the same time, the risk of calcium oxalate stone formation increases with the number of features of the MS [21]. 

The role of FR as a causal factor of urolithiasis is still puzzling. On the one hand, epidemiological and clinical studies have shown that FR may be responsible for this disease, while on the other hand consumption of fruits and juices in a diet is still recommended for patients with kidney stones [22]. To our knowledge, the studies conducted so far have not directly confirmed the relationship between the FR diet and the presence of renal calculi. Therefore, we utilized a protocol with diets differing in FR content and its form (liquid vs. solid) to evaluate the kidney tubule function and markers of MS. An experimental model has been utilized to perform a simultaneous histopathological assessment for kidney stones.

## 2. Results

The results are expressed as mean ± SD and shown in the following tables.

### 2.1. Assessment of Nutritional and Metabolic Status

The serum FR concentration showed tendency to higher values in rats on F10 and F60 diet compared to RD, with the highest level in F10 group. This may indicate that FR in a liquid form is more easily absorbed from the gastrointestinal tract than in a solid form. The rats showed no difference in total daily calorie intake per day at the end of the experiment, although animals on the F10 and F60 diet consumed significantly less chow than those in the RD group. This was probably due to proportionally more calories consumed in the form of FR. The rats on the F60 diet had significantly lower weight gain, which might be indicative of its toxic potential. There was also a tendency for a lower albumin concentration in serum on the fructose-rich diet. Creatinine concentration was lower in F60 as compared to F10 due to a lower body weight of the animals at the end of the experiment. The rats on the F10 and F60 diet, compared to the RD group, showed a trend towards lower values of creatinine clearance. After eight weeks on a fructose-rich diet, the rats presented some features of the metabolic syndrome, namely there was a tendency for higher serum concentration of triglycerides, insulin and high score for the homeostasis model assessment for insulin resistance (HOMA). Interestingly, compared to F60 (solid), the F10 diet (fluid) resulted in a higher insulin concentration in serum. There were no differences with regard to UA levels in serum. The homocysteine (HCS) level significantly increased with the higher FR content in diet. A tendency for higher erythropoietin (EPO) levels was observed in F10 and F60 compared to RD. There were no significant differences in serum concentrations of sodium (Na), potassium (K), magnesium (Mg) and calcium (Ca) despite a tendency for higher phosphate level (Pi) and higher calcium-phosphate product (CaxPi) in fructose-rich diet (Table 1).

### 2.2. Assessment of Urine 

The rats from the F10 group that were consuming FR as a 10% water solution, tended to drink more water and produced twice as much urine as rats in the RD and F60 groups. The urine pH decreased (denoting higher acidity) as the FR content in the diet increased, and it was significantly lower in the F60 group compared to RD and F10 groups. However, the specific gravity increased with the content of FR in the diet and was significantly higher in the F60 group. There were no differences in proteinuria between the groups. Significantly higher urinary excretion of monocyte chemoattractant protein-1 (MCP-1) and N-acetyl-ß-D-glucosaminidase (NAG) per urinary creatinine was shown with increasing FR content in the diet. Urinary uric acid excretion (uUAEx) decreased with increasing FR content in the diet and was significantly lower in the F10 and F60 groups compared to RD. However, the levels of uric acid clearance (UACl) and fractional uric acid excretion (UAFEx) did not differ between groups. An FR-rich diet resulted in a significantly higher urinary clearance, as well as total excretion and fractional excretion of phosphorus, sodium and calcium. The F60 diet was associated with increased phosphaturia (>30×) and natriuria (4×) and, to a lesser extent, with increased calciuria. On the other hand, the F10 diet mainly led to increased calciuria (3×), and, to a lesser extent, to increased phosphaturia and natriuria. There were no statistical differences in potassium and magnesium excretion with urine between the groups (Table 2). 

### 2.3. Analysis of Urinary Electrolytes Excretion as a Percentage of Dietary Intake 

Due to the differences in the composition of the F60 diet as compared to the diet used in the RD and F10 groups, a comparative analysis of the amount of sodium, calcium and phosphate provided in each diet was performed. Dietary sodium intake from diet was 20% lower in F10 and 20% higher in F60 compared to the RD. Calcium and phosphate intake from diet was lower for both electrolytes by 30% in F10 than RD. In the same time, it was reducing by 60% for calcium and 30% for phosphate in F60 as compared to RD. The ratio of urinary electrolytes (sodium, calcium, phosphate) excretion as a percentage of dietary intake was also calculated (Table 3).

### 2.4. Kidney Histopathology 

The microscopic examination of kidney specimens revealed a focal interstitial and tubular basophilic deposits (calcium phosphate stones) in two animals from the F10 group and another two animals from the F60 group. The stones were located in the cortex and renal medulla, in all parts of the nephron. In haematoxylin and eosin (HE) stained tissue sections the concretions were non polarizable, small, round and exhibiting psammoma body-like substructure with concentric rings (Figure 1, Figure 2, Figure 3 and Figure 4). Von Kossa staining revealed brown-black calcifications. Additionally, the light microscopy revealed focal and mild tubulo-interstitial lymphocytic infiltration in rats from F10 and F60 groups (Figure 5 and Figure 6). There were no morphological features of acute tubular injury, inflammation nor calcium depositis in kidney of rats from RD group (Figure 7).

## 3. Discussion

Our study confirmed that FR intake led to the development of some features of MS as an increased serum insulin, HOMA and triglyceride level. FR also induced local inflammatory response, mediated by MCP-1, leading to the renal proximal tubular injury, indicated by increased urinary excretion of NAG and the presence of mononuclear cells infiltration in kidney parenchyma (Figure 5 and Figure 6). Interestingly, our study also confirmed that FR given in the form of fluid (F10) as compared to a solid feed (F60) resulted in a higher insulin concentration in serum. This was probably due to the fact that F10 was more eagerly consumed by animals than F60 (Table 1). In our previous study conducted on the same group of rats, we showed that F10 versus F60 stimulated higher secretion of insulin and C-peptide by the pancreatic islet cells, and lead to more pronounced insulin resistance [9]. Moreover, our study showed that FR given as a 10% liquid solution significantly increased uCaEx and reduced uUAEx, while in the form of a solid 60% feed, it further decreased the uUAEx, led to urine acidification and induced severe uPiEx (Table 2). 

An important observation of this study was a microscopic confirmation of precipitated calcium deposits in the renal tubules and kidney parenchyma only in rats on high-fructose diet, respectively, in two rats from F10 (Figure 1 and Figure 2) and another two in F60 groups (Figure 3 and Figure 4). Based on histopathological analysis (HE and von Kossa staining) we assumed that those concretions can be made of calcium phosphate (Figure 1, Figure 2, Figure 3 and Figure 4). A stone forming rats from F10 and F60 groups were characterized by a significantly higher concentration of serum insulin, increased CaCl, as well as increased uCaEx, CaFEx, and, interestingly, increased uMgEx (Table 4). Interestingly, due to the fact that F10 rats, compared to F60 rats, excreted twice as much urine of low specific gravity, our results suggest that despite hypersaturation of urine, there are also other specific metabolic disturbances induced by FR responsible for lithogenesis. 

The commonly crystallizing calcium salts in the kidney are calcium phosphate (CaPi) and calcium oxalate (CaOx) [22], the latter can form on the basis of UA crystals [19]. It is thought that high urine pH favours calcium phosphate (CaPi) crystallization, while low or neutral pH promoted precipitation of calcium oxalate (CaOx) [23]. Studies on animal model indicate that intratubular nephrocalcinosis is a result of urinary supersaturation with a variety of compounds including CaPi, CaOx and UA, while interstitial nephrocalcinosis is associated with inflammation and the deposition of poorly crystallized CaPi [24]. Furthermore, CaPi crystals can precipitate in renal tubules and form extends outward through kidney interstitium to the renal papilla where promote crystallization of CaOx stones [23].

### 3.1. Fructose Induced Metabolic Syndrome and Inflammation

It has been confirmed in a rat model that a fructose-enriched diet given for 5 weeks may induce a complete MS, including hypertension, hyperinsulinemia, hypertriglyceridemia and hyperhomocysteinemia [25]. HCS stimulates MCP-1 expression in the renal proximal tubular cells by activation of nuclear factor-kappa B (NF-κB) and may contribute to progression of chronic kidney disease [26]. 

We found that serum concentration of HCS and urinary excretion of MCP-1 and NAG increased with FR content in diet, reaching a statistical significance for the F60 group compared to the RD group (Table 2). While the impact of an FR-rich diet on inflammatory markers and proximal tubular injury was proportional to the FR content in diet, it is suggested that the toxic effect might depend on the FR concentration in the tubular fluid. It has also been confirmed that FR increases intracellular levels of UA in the proximal tubular cells, which further induces an inflammatory response [27]. Similarly, in the study conducted on rats receiving 250 g/L of FR in the form of a fluid solution for 16 months versus glucose and sucrose diet, it has been established that the inflammatory injury of the renal tubules and glomeruli are proportional to the FR concentration in urine [28]. Additionally, ingestion of 10% FR for eight weeks has induced renal microvascular disease and glomerular hypertension in rats [12], while the diet containing 60% of FR versus normal or dextrose diet caused tubule-interstitial injury and interstitial macrophage infiltration in rats [29,30]. 

In fact, we found a tendency for higher concentration of EPO in the F10 and F60 groups compared to the RD group, which might be a hallmark of tubule-interstitial injury secondary to inflammation or ischemia [31]. Meanwhile, the microscopic examination confirmed that both F10 and F60 rats showed the presence of focal lymphocytic infiltrates within the tubulointerstitial compartment (Figure 5 and Figure 6). However, an FR diet for an 8-week period was not sufficient to produce a significant change in creatinine clearance (CrCl) between F10, F60 and RD groups (Table 1). Undoubtedly, extending the duration of fructose-rich diet period would have led to a more significant kidney injury. 

Insulin resistance acts at the kidney level by reducing urinary pH, thus hampering the ability of the kidney to generate renal ammonium in response to an acid load [32]. Furthermore, insulin resistance is characterized by increased excretion of titratable acids (e.g., phosphates and sulphates) and low urinary concentration of citrate [19]. The other authors have also reported lower uUAEx in fructose-fed rats [31,33]. This phenomenon in rats fed on a 10% FR diet can rely on an abnormal expressions of urate transporters in the proximal tubule secondary to hyperinsulinemia, dyslipidaemia and inflammation [33]. Furthermore, a fructose-rich diet given for three months does not lead to significant differences in serum UA concentration, and eventually prolongation of the diet up to 5 months has resulted in a significant hyperuricemia [17]. It is also thought that FR does not increase uric acid level significantly in rats due to the presence of the uricase which causes degradation of uric acid [10]. Similarly to previously mentioned studies, the significantly lower uUAEx was present in both F10 and F60 groups as compared to the RD group, despite the fact that there was no significant differences in concentration of UA in serum, nor UACl and UAFEx (Table 1 and Table 2). Moreover, patients with MS are also characterized by increased uCaEx and the risk of calcium stone formation in such patients is proportional to the number of features of the MS [21]. Those factors may predispose to uric acid and calcium stones formation, as both low urine pH and low urine volume are known risk factors of UA precipitation [32]. 

### 3.2. Fructose Effects on Macro-Mineral Homeostasis

Our study confirmed that an FR diet negatively affects macro-mineral homeostasis leading to significantly higher uCaEx, as well as uPiEx and uNaEx in relation to their dietary intake (Table 3). The same observation has also been shown in humans [34]. Although we did not find a significant difference in serum calcium (Ca) level, a significantly higher uCaEx was present in the F10 group compared to the RD and F60 groups. However, in the F60 group, uCaEx was two times smaller as compared to the F10 group and was only insignificantly higher compared to the RD group (Table 2). This fact was not related to the diet, since the content of Ca consumed by rats was significantly lower in groups F10 and F60 compared to the RD group. At the same time, the ratio of uCaEx per Ca consumed from the diet was significantly higher in groups F10 and F60 than in RD group (Table 3).

Up to 60% of filtered Ca is reabsorbed in the proximal tubule due to the sodium–hydrogen exchanger 3 (NHE3), which is responsible for trans-epithelial sodium flux and consequently provides the osmotic driving force for water flux and the driving force for passive para-cellular Ca flux [35]. NHE3 is also known to be regulated by the calciotropic hormones, i.e., PTH [36] and 1,25-dihydroxy vitamin D_3_ [37]. It has been confirmed that an increase of uCaEx is characteristic for all sugars including glucose and sucrose [38]. 

An FR diet, as compared to glucose, can increase Ca content in the rat kidney homogenates up to eight times, but only when the diet is lacking in magnesium [16]. We did not find significant differences in serum Mg or uMgEx in a baseline characteristics of the groups (Table 1 and Table 2). Although, the stone-forming rats on the FR diet were characterized by significantly higher uMgEx than rats on the FR diet without deposits in kidney on microscopy (Table 4). Mg is generally known to inhibit lithogenesis as it can bind to oxalate [39] and it competes with Ca ions to make a complex with oxalate, which is more soluble than CaOx [40]. There have also been reports discrediting any beneficial effects of Mg therapy for nephrolithiasis [41]. On the other hand, the Mg-deficient rats on FR diet have developed nephrocalcinosis caused by deposition of CaPi rather than CaOx [16]. Therefore, the effect of Mg on the development of CaPi kidney stones may be limited. Due to technical problems, we did not measure oxalate excretion in urine and thus its role in the formation of calcium deposits in this experiment could not be ruled out. 

At the same time, the uPiEx and uNaEx increased along with the FR content in diet in our study. The uPiEx was two times greater in the F10 group and more than thirty times higher in the F60 group compared to the RD group (Table 2). However, only a tendency to higher serum Pi concentration in the F60 group was observed as compared to the F10 and RD groups (Table 1). The rats in F10 and F60 groups consumed about 30% less of Pi from diet compared to the RD group. Despite this fact, the ratio of uPiEx to Pi consumed from diet was severely elevated, especially in F60 rats (Table 3). The other studies have also confirmed that dietary FR, as compared to glucose [42] and starch [35], results in greater uPiEx in rats [42]. Moreover, a study on rats fed 63% FR confirms a significant decrease in Ca and Pi content in the femur and tibia [43]. Thus, FR-rich diet has also led to a negative Pi balance secondary to increased serum alkaline phosphatase activity and higher uPiEx [35]. Research showed that 60–70% of the filtered Pi is reabsorbed in the proximal tubule [44]. Pi reabsorption is achieved mainly by controlling the apical expression of a few types of sodium-phosphate cotransporters (NaPi) localized in renal proximal tubules and are regulated by PTH, vitamin D and dietary Pi [37]. 

In order to understand the mechanism of increased calcium and phosphorus excretion the concentration of PTH, FGF-23 and 25(OH)-vitamin D3 (D3) was measured. There were no statistically significant differences in the analyzed parameters between the groups. Although, there was a trend towards a higher concentration of PTH and a lower level of D3 in F10 and F60 groups compared to the RD group (Table 1). Therefore, the results of our experiment may indicate renal tubular damage associated with a high-fructose diet. 

Interestingly, a study exploring the effect of ingestion of 200 g of FR daily for two weeks by healthy men has also confirmed an increase in PTH concentration and a decrease in serum ionized calcium [45]. meanwhile, another study conducted on rats fed with FR for four weeks indicated a decrease in circulating levels of 1,25-(OH)2D3 due to diminished activity of 1α-hydroxylase with secondary inhibition of calcium reabsorption in the intestines and kidneys. Interestingly, those animals also developed osteoporosis [46]. 

Fructose-induced tubulointerstitial injury may be connected with secondary deficiency of active vitamin D3 and disturbances in PTH production. Those abnormalities eventually might be responsible for the progression of CKD and have a detrimental effect on bones. Thus, fructose emerges as a culprit responsible not only for today’s epidemic of cardiovascular diseases but also for nephrolithiasis and osteoporosis. 

Finally, we would like to mention some limitations of the current study. The composition of the diet in the F60 group was different, apart from the fructose, in respect to protein, fat, vitamin D, and mineral content compared to F10 and RD groups. The choice of this feed was determined by the fact that it was the only commercially available diet with 60% fructose content. Since the composition of the F60 diet could affect the results of urinary excretion of calcium and phosphate, the amount of food and minerals consumed by the animals was analyzed during the experiment. An evaluation of urinary excretion of oxalate and citrate should be performed, as data from the literature suggest that the FR diet and metabolic syndrome may be associated with increased oxalate excretion in urine and decreased urinary citrate excretion. The present study was conducted on a small groups of male rats. Due to the fact that sex hormones may influence the urine composition, the effect of FR-diet should also be confirmed in female rats. It would also be appropriate to use a technique which does not rinse out any uric acid crystals during tissue preparation for histopathology evaluation. Application of X-ray diffraction analysis technique for precise characterization of the intraluminal and interstitial deposits would have added highly valuable information to this study. 

Despite those shortcomings, we believe that the preliminary results of this study provide valuable information as they confirm, for the first time, the direct relationship between the high-FR diet and the presence of precipitated calcium deposits in the renal tubules and kidney parenchyma confirmed by microscopy examination.

## 4. Materials and Methods

### 4.1. Animal Model

All animal procedures were performed in compliance with the National Institutes of Health Guide for the Care and Use of Laboratory Animals and approved by the Institutional Committee for Animal Welfare at Bydgoszcz (no. 13/2007 and 22/2011), in accordance with the Polish Act on Experiments on Animals (Journal of Laws 2005/33/289) and the Directive of the European Parliament No. 2003/65/EC. Male Wistar rats were purchased from Mossakowski Medical Research Centre, the Polish Academy of Sciences, Warsaw, Poland. The animals had the initial body mass of 386 ± 40 g and were at the age of eight weeks at the beginning of the experiment. They were randomly assigned to three groups and kept for eight weeks on the following diets: RD (*n* = 5) regular diet with FR concentration < 3%, given in the form of pelleted chow (Agropol, Poland); F10 (*n* = 6) consisting of regular diet supplemented with 10% solution of FR in drinking water (Biofan Fruktoza, Poland); and F60 (*n* = 5) consisting of 60% FR in diet in the form of pelleted chow (TD.89247, Harlan Tekland, Cambridge, WI, USA). The RD diet contained 16% protein, 2.8% fat, 1.10% calcium, 0.7% phosphorus, 0.22% sodium, and 800 IU/kg of vitamin D3. The F60 diet contained 18.3% protein, 5.2% fat, 0.6% calcium, 0.54% phosphorus, 0.49% sodium, 500 IU/kg of vitamin D3. The animals were maintained under a constant 12 h photoperiod in a temperature between 21 °C and 23 °C. They were allowed free access to water and chow. The daily energy intake was calculated. The body mass was measured at the beginning and at the end of the experiment. Furthermore, at the end of the study, 24 h urine collections were conducted and samples were collected using individualized metabolic cages. The rats were then sacrificed and blood samples were collected from the right ventricle of the heart for biochemical analysis following a 4 h period of fasting. The urine and serum were collected and stored at −80 °C before further analysis.

### 4.2. Laboratory Tests

Serum BUN, creatinine (Cr), albumin, glucose, triglycerides, uric acid (UA), sodium (Na), potassium (K), calcium (Ca), phosphates (Pi) and magnesium (Mg) were measured using Architect ci8200 (Abbott Laboratories, Wiesbaden, Germany) according to the manufacturer’s operating protocol. Serum fructose (FR) concentration was determined based on the quantitative colorimetric method (BioAssay Systems, Hayward, CA 94545, USA). Serum insulin was tested using the ELISA assay (Alpco Immunoassays, Salem, NH 03079, USA). The homeostasis model assessment for insulin resistance (HOMA-IR) was calculated using the following equation: [fasting plasma insulin (ng/mL) x fasting plasma glucose (mg/dL)/405]. EPO was determined using the ELISA assay (Roche Diagnostics GmbH, Mannheim, Germany).

Serum homocysteine was determined by the fluorescence polarization immunoassay using AxSYM (Abbott Laboratories, Wiesbaden, Germany). Quantitative determination of parathyroid hormone (PTH) and fibroblast growth factor 23 (FGF-23) level in plasma was done by ELISA assays (Immutopics International, San Clemente, CA 92673, USA). Serum concentration of 25(OH) Vitamin D_3_ was tested using the Elisa Kit (Abcam, Cambridge, MA, USA). The analysis of a 24 h urine collection included measurement of the volume, urine pH, specific gravity, as well as daily excretion of protein, monocyte chemoattractant protein-1 (MCP-1), and N-acetyl-β-D-glucosaminidase (NAG). Urine Cr, UA, P, Ca, Mg, Na and K were measured using Architect ci8200 (Abbott Laboratories, Wiesbaden, Germany). The urine specific gravity and pH were determined by urine dipstick test (Combur-10 Test; Urisys 1100; Cobas, Roche Diagnostics GmbH, Mannheim, Germany). MCP-1 in urine was tested with the ELISA assay (IBL America, Minneapolis, MN 55432, USA), while NAG in urine was tested by colorimetry (Roche Diagnostics GmbH, Mannheim, Germany). Renal clearances (Cl) as well as fractional excretions (FEx) of Cr, UA, P, Ca, Mg, Na and K were calculated using the following formulas:

Cl_(x)_ (mL/min) = [U_(x)_ (mg/dL) × urine volume (mL/24h)]/[S_(x)_ (mg/dL) × 1440 (min)]. FEx_(x)_ = (U_(x)_ × S_Cr_/U_Cr_ × S_(x)_) × 100

where U is the concentration of the analyzed substance (x) in urine and S is the concentration of the analyzed substance in serum. 

### 4.3. Histology

The kidneys were removed, decapsulated, placed in 10% buffered formalin and prepared for light microscopy (LM). The 5 µm thick paraffin-embedded sections were stained with hematoxylin and eosin as well as von Kossa stain, and later studied using bright field and polarized microscopy. The specimens were assessed for presence of pathological lesions, especially in the tubulointerstitial compartment, as well as for crystal distribution within the kidneys. Oxalate crystals are typically translucent, whereas calcium phosphate crystals often appear blue or purple on H&E staining. The von Kossa stain reacts with the phosphate anion of calcium phosphate and, therefore, does not stain calcium oxalate [47].

### 4.4. Statistical Analysis

Statistical analysis was conducted using the one-way analysis of variance with Bonferoni post-hoc test. The differences were considered statistically significant for *p* < 0.05.

## 5. Conclusions

In summary, we showed that dietary fructose consumption is associated with induction of the metabolic syndrome phenotype and local inflammatory response (MCP-1) in the proximal tubules. The defect of the proximal tubules can be observed even at as low as 10% fructose content in diet. While a 60% fructose diet exerts a more toxic effect on the renal tubule function. Those disturbances lead to defects in uric acid excretion, as well as induction of hypercalciuria and hyperphosphaturia. Thus, fructose-induced metabolic syndrome together with supersaturation of urine with mineral salts lead to precipitation of calcium salts in the renal tubules, while concurrent inflammation predisposes to interstitial nephrocalcinosis. An interesting observation of this work is the fact that both of the tested high-fructose diets, given for an 8-week period, based on a liquid 10% FR solution or a solid 60% FR feed, were able to induce urolithiasis in rats characterized by high insulin level and hypercalciuria. Fructose-induced tubulo-interstitial injury may be connected with secondary deficiency of active vitamin D3 and disturbances in PTH production.

## Figures and Tables

**Figure 1 ijms-23-00203-f001:**
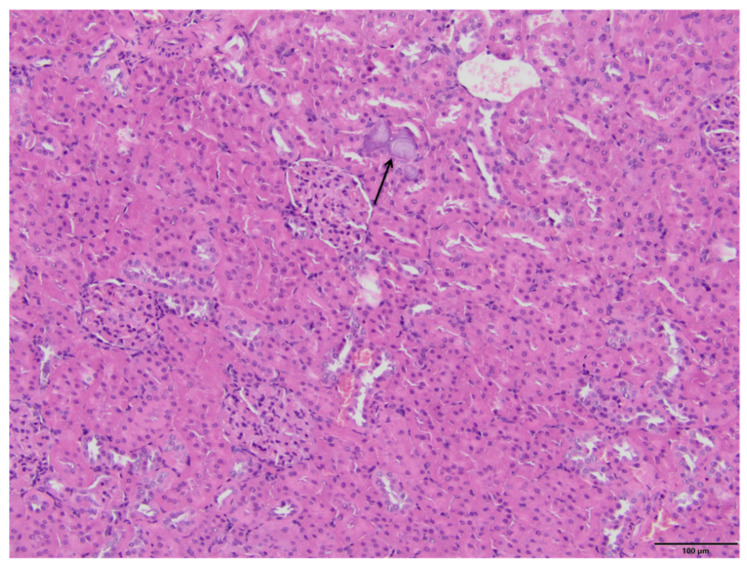
Pathological lesions in kidneys of rats on 10% fructose diet (F10). The purple concretions within the interstitium and tubules with ring-like substructures (arrow). The deposits are non-polarizable, displacing the proximal tubule epithelium and narrowing the lumen. (HE, magnification 20×).

**Figure 2 ijms-23-00203-f002:**
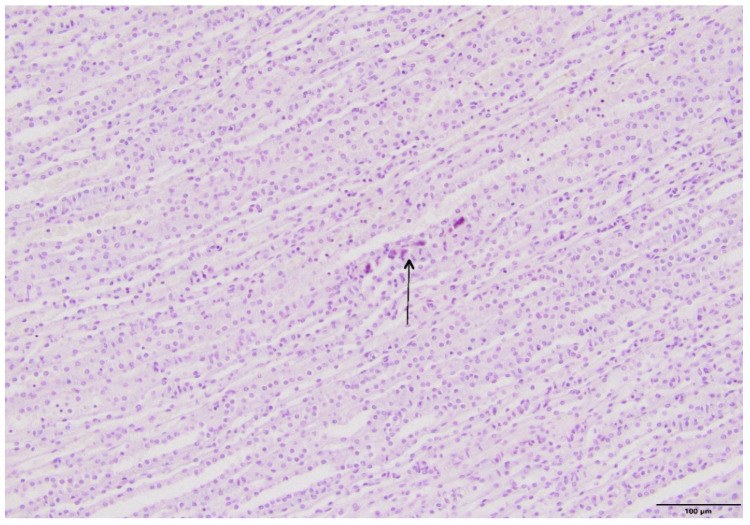
Pathological lesions in kidneys of rats on 10% fructose diet (F10). Basophilic deposits in area of renal medulla (arrow) (HE, magnification 20×).

**Figure 3 ijms-23-00203-f003:**
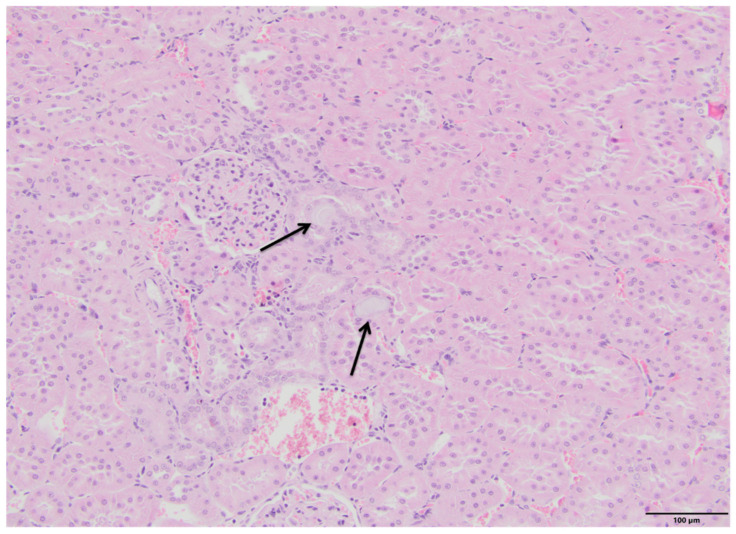
Pathological lesions in kidneys of rats on 60% fructose diet (F10). Two pale basophilic deposits with delicate ring-like substructure within interstitium (arrows). The deposits are non-polarizable, compressed the tubular structures. (HE, magnification 20×).

**Figure 4 ijms-23-00203-f004:**
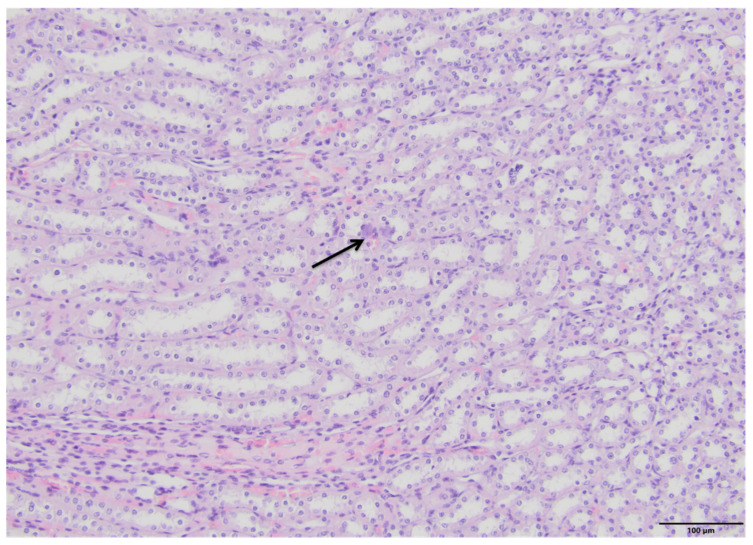
Pathological lesions in kidneys of rats on 60% fructose diet (F60). Two small basophilic interstitial and tubule-associated concretions in renal medulla (arrow). (HE, magnification 20×).

**Figure 5 ijms-23-00203-f005:**
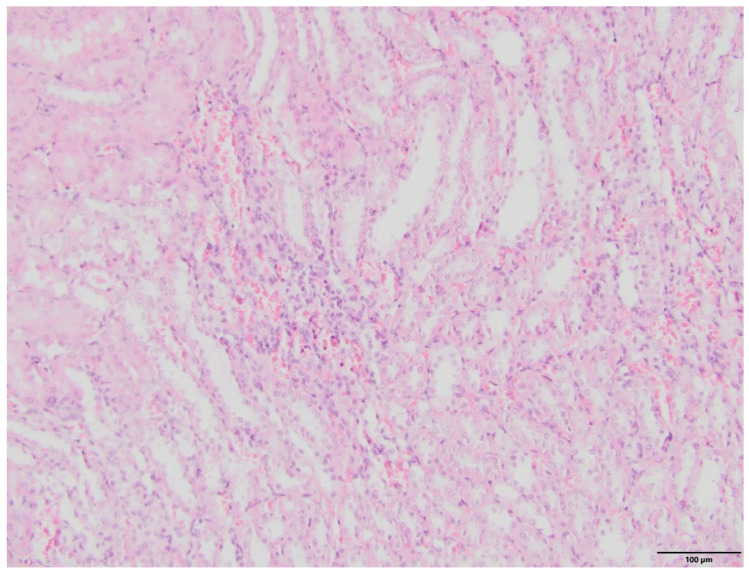
Pathological lesions in kidneys of rats on 10% fructose diet (F10). Mild tubule-interstitial mononuclear cell infiltration. (HE, magnification 20×).

**Figure 6 ijms-23-00203-f006:**
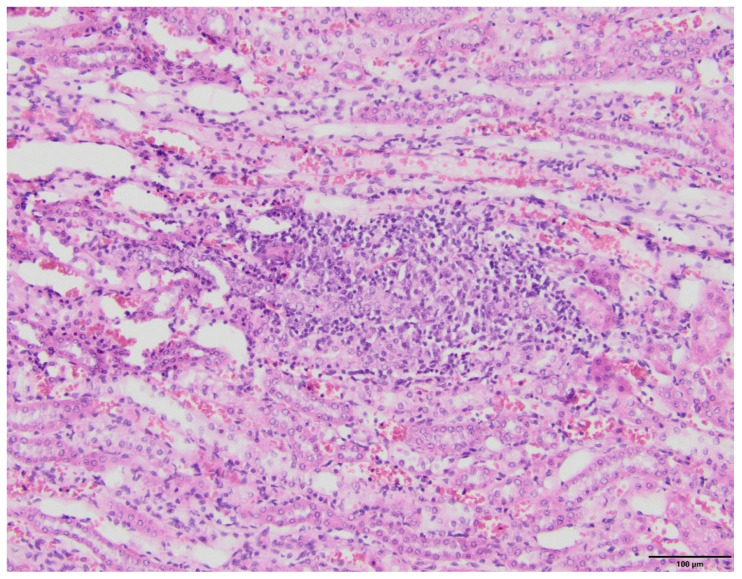
Pathological lesions in kidneys of rats on 60% FR diet (F60). Moderate lymphocytic infiltration of renal tubules and interstitium. (HE, magnification 20×).

**Figure 7 ijms-23-00203-f007:**
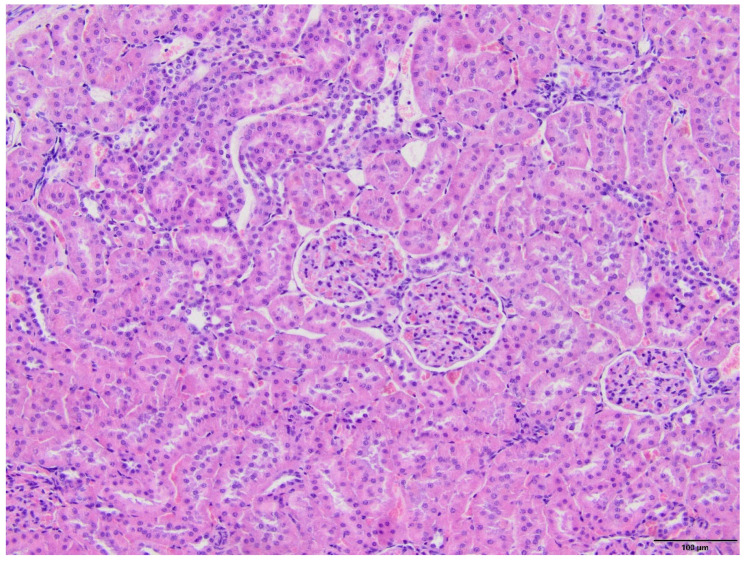
Kidney of rat on regular diet (RD). All structural elements of the kidney are seen: three glomeruli without structural abnormalities, normal tubules, interstitium and blood vessels. (HE, magnification 20×).

**Table 1 ijms-23-00203-t001:** Nutritional status and serum laboratory data of the animals after 8 weeks period on a regular diet (RD), a diet supplemented with 10% of fructose (F10) and 60% of fructose (F60).

Determined Parameters	RD (I)	F10 (II)	F60 (III)	ANOVA	P
Food consumption [g/day]	37.9 ± 2.5	27.2 ± 4.5	28.6 ± 2.4	<0.001	I vs. II I vs. III
Water consumption [mL/day]	56 ± 9	90 ± 24.5	50 ± 14	<0.01	I vs. II II vs. III
Weight gain [gram]	275 ± 23	275 ± 30	219 ± 37	<0.01	I vs. III II vs. III
Total energy value [kcal per day]	99.6 ± 6.6	96.6 ± 10.3	102.8 ± 8.9	NS	
Energy from fructose [kcal/day]	3 ± 0.2	27.2 ± 5.5	61.7 ± 5.3	<0.001	I vs. II I vs. III II vs. III
Albumin [g/dL]	3.3 ± 0.1	3.2 ± 0.2	3.1 ± 0.2	NS	-
Fructose [mg/dL]	0.59 ± 0.05	1.26 ± 0.7	1.04 ± 0.5	NS	-
BUN [mg/dL]	20.8 ± 1.4	18.1 ± 5.2	18.7 ± 2.6	NS	-
HCS [µmol/L]	4.5 ± 0.2	5.8 ± 1.4	6.4 ± 1.5	< 0.05	I vs. III
Creatinine [mg/dL]	0.54 ± 0.04	0.59 ± 0.09	0.47 ± 0.04	< 0.05	II vs. III
Creatinine clearance [mL/min/100 g]	0.47 ± 0.08	0.40 ± 0.08	0.44 ± 0.09	NS	-
Erythropoietin [mIU/mL]	0.65 ± 0.56	1.72 ± 2.01	1.33 ± 0.84	NS	-
Uric Acid [mg/dL]	1.76 ± 0.4	1.45 ± 0.3	1.56 ± 0.1	NS	-
Insulin [ng/mL]	3.87 ± 2.0	5.54 ± 2.4	5.31 ± 1.2	NS	-
HOMA [AU]	1.94 ± 0.95	3.13 ± 1.036	2.36 ± 0.81	NS	-
Triglicerides [mg/dL]	158 ± 34	180 ± 27	221 ± 83	NS	-
Cholesterol [mg/dL]	74 ± 11	60 ± 9	66 ± 9	NS	-
PTH [pg/mL]	225 ± 112	283 ± 136	492 ± 340	NS	-
Vitamin 25(OH)D3 [ng/mL]	52 ± 11	19 ± 17	31 ±44	NS	-
FGF-23 [pg/mL]	212 ± 66	379 ±324	216 ± 51	NS	-
Calcium [mmol/L]	2.52 ± 0.05	2.53 ± 0.11	2.44 ± 0.09	NS	-
Phosphate [mmol/L]	2.32 ± 0.1	2.39 ± 0.2	2.60 ± 0.2	NS	-
CaxPi [mmol^2^/L^2^]	5.85 ± 0.36	6.04 ± 0.49	6.34 ± 0.59	NS	-
Magnesium [mmol/L]	2.26 ± 0.1	2.54 ± 0.4	2.23 ± 0.3	NS	-
Sodium [mmol/L]	140.02 ± 1.68	139.34 ± 3.50	140.03 ± 2.54	NS	-
Potassium [mmol/L]	5.14 ± 0.95	5.42 ± 0.72	5.07 ± 0.48	NS	-

BUN—blood urea nitrogen; CaxPi—product of calcium × phosphate; HCS—homocysteine; HOMA—homeostasis model assessment for insulin resistance; NS—non-statistically significant; PTH—parathyroid hormone; FGF-23—fibroblast growth factor 23; NS—non-statistically significant.

**Table 2 ijms-23-00203-t002:** The results of urinalysis and markers of kidney tubular function after 8 weeks period on a regular diet (RD), a diet supplemented with 10% of fructose (F10) and 60% of fructose (F60).

Determined Parameters	RD (I)	F10 (II)	F60 (III)	ANOVA	P
Urine output [mL/day]	26 ± 9.6	56 ± 27	25 ± 13	< 0.001	I vs. II II vs. III
Urine pH	8.5 ± 0.57	8.2 ± 0.83	5.6 ± 0.51	< 0.001	I vs. III II vs. III
Urine specific gravity [g/L]	1.0125 ± 0.003	1.014 ± 0.004	1.026 ± 0.004	< 0.001	I vs. III II vs. III
PCR [mg/mg Cr]	1.0 ± 0.4	1.1 ± 0.5	0.8 ± 0.5	NS	-
MCP-1/Cr [ng/mg Cr]	3.5 ± 0.6	4.4 ± 4.0	11.2 ± 2.5	< 0.01	I vs. III II vs. III
NAG/Cr [U/g Cr]	8.6 ± 5	15.1 ± 7.6	20.8 ± 5	< 0.05	I vs. III
uUAEx [mg/day]	3.28 ± 0.51	2.58 ± 0.57	2.18 ± 0.66	< 0.05	I vs. III
UACl (mL/min)	0.10 ± 0.04	0.13 ± 0.04	0.10 ± 0.03	NS	-
UAFEx [%]	4.4 ± 1.2	5.2 ± 1.6	4.1 ± 1.6	NS	-
uNaEx [mg/day]	33.3 ± 8.5	40.9 ± 10.8	166.8 ± 27.2	< 0.001	I vs. III II vs. III
NaCl [mL/min]	0.007 ± 0.002	0.008 ± 0.002	0.031 ± 0.013	< 0.001	I vs. III II vs. III
NaFEx [%]	0.24 ± 0.08	0.34 ± 0.06	1.16 ± 0.45	< 0.001	I vs. III II vs. III
uPiEx [mg/day]	1.85 ± 1.61	4.42 ± 5.14	75.30 ± 21.95	< 0.001	I vs. III II vs. III
PiCl [mL/min]	0.017 ± 0.015	0.044 ± 0.056	0.567 ± 0.284	< 0.001	I vs. III II vs. III
PiFEx [%]	0.6 ± 0.6	1.7 ± 0.2	21 ± 5	< 0.001	I vs. III II vs. III
uCaEx [mg/day]	2.02 ± 0.89	5.71 ± 3.05	3.08 ± 1.92	< 0.05	I vs. II
CaCl [mL/min]	0.014 ± 0.006	0.038 ± 0.020	0.019 ± 0.014	< 0.05	I vs. II
CaFEx [%]	0.45 ± 0.2	1.46 ± 0.7	0.75 ± 0.5	< 0.05	I vs. II
uMgEx [mg/day]	5.2 ± 1.9	5.9 ± 2.1	3.9 ± 1.4	NS	-
MgCl [mL/min]	0.19 ± 0	0.17 ± 0.07	0.11 ± 0.3	NS	-
MgFEx [%]	5.4 ± 0	6.7 ± 2.5	4.4 ± 0.7	NS	-
uKEx [mg/day]	174.3 ± 19.9	152.1 ± 26.1	124.9 ± 45.3	NS	-
KCl [mL/min]	0.61 ± 0.1	0.51 ± 0.1	0.45 ± 0.17	NS	-
KFEx [%]	19.74 ± 2.15	19.92 ± 4.73	16.85 ± 5.46	NS	-

uCaEx—urinary calcium excretion; CaCl—calcium clearance; CaFEx—calcium fractional excretion; UACl—uric acid clearance; uUAEx—urinary uric acid excretion; UAFEx—uric acid fractional excretion; uNaEx –urinary sodium excretion; NaCl—sodium clearance; NaFEx—sodium fractional excretion; uPiEx—urinary phosphate excretion; PiCl—phosphate clearance; PiFEx—phosphate fractional excretion; uMgEx—urinary magnesium excretion; MgCl—magnesium clearance; MgFEx—magnesium fractional excretion; uKEx—urinary potassium excretion; KCl—potassium clearance; KFEx—potassium fractional excretion; MCP-1/Cr—urinary monocyte chemoattractant protein-1 per urinary creatinine ratio; NAG/creatinine ratio—urinary N-acetyl-ß-D-glucosaminidase per urinary creatinine ratio; PCR—urinary protein to creatinine ratio; NS—non-statistically significant.

**Table 3 ijms-23-00203-t003:** The mean daily amount of Sodium, Calcium and Phosphate provided in RD, F10 and F60 diet and the ratio of daily urinary Excretion of Sodium, Calcium and Phosphate as percentage of their daily dietary intake from diet.

Determined Parameters	RD (I)	F10 (II)	F60 (III)	ANOVA	P
Sodium intake from diet [mg/day]	113.7 ± 7.6	81.6 ± 13.6	142.0 ± 12.4	<0.001	I vs. II I vs. III II vs. III
uNaEx/Na consumed in diet [%]	29.2 ± 6.8	50.7 ± 12.7	117.0 ± 10.4	<0.001	I vs. II I vs. III II vs. III
Calcium intake from diet [mg/day]	417 ± 28	299 ± 50	174 ± 15	<0.001	I vs. II I vs. III II vs. III
uCaEx/Ca consumed in diet [%]	0.5 ± 0.2	1.9 ± 0.9	1.8 ± 1.1	<0.05	I vs. II I vs. III
Phosphate intake from diet [mg/day]	227 ± 15	163 ± 27	156 ± 14	<0.001	I vs. II I vs. III
uPiEx/Pi consumed in diet [%]	0.8 ± 0.7	3.0 ± 3.6	47.7 ± 11.1	<0.001	I vs. III II vs. III

Ca—calcium; Na—sodium; Pi—phosphate; uCaEx—urianry calcium excretion; uNaEx—urinary sodium excretion; uPiEx—urinary phosphate excretion.

**Table 4 ijms-23-00203-t004:** Metabolic characteristics of rats on fructose-rich diet (F10 + F60) developing stones versus animals without kidney stones.

Metabolic Characteristics	Rats without Kidney Deposits (N = 7)	Rats with Kidney Deposits (N = 4)	ANOVA
Insulin [ng/mL]	4.60 ± 1.60	6.86 ± 0.83	<0.05
uCaEx [mg/day]	2.02 ± 0.89	5.71 ± 3.05	<0.01
CaCl [mL/min]	0.018 ± 0.011	0.047 ± 0.015	<0.01
CaFEx [%]	0.73 ± 0.42	1.67 ± 0.58	<0.05
uMgEx [mg/day]	3.93 ± 1.22	6.39 ± 2.16	<0.05

CaCl—calcium clearance; uCaEx—urinary calcium excretion; CaFEx—calcium fractional excretion; uMgEx—urinary magnesium excretion.

## Data Availability

The data presented in this study are available on request from the corresponding author.

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
