# Peer review of "Fructose-Rich Diet Is a Risk Factor for Metabolic Syndrome, Proximal Tubule Injury and Urolithiasis in Rats"

_ijms, 2021, doi:10.3390/ijms23010203_

Round 1
Reviewer 1 Report
Authors presented the impact of high-fructose content diet on metabolic syndrome and mineral homeostasis. Although the negative influence of fructose on metabolism is already commonly known, I have read the paper by Flisinski et all. with great interest.
In my opinion the paper is well written. The experimental design is properly planed, and statistical analysis is conducted accordingly. The paper proves a toxicity related to high consumption of fructose and especially in case of fluid intake, what is of great importance in the world were a glass of fruit juice is presented as a symbol of health.
I do not have other remarks
Author Response
We would like to thank you very much for your review of our article.Reviewer 2 Report
Reviewer’s Comment
In the research article titled ‘Fructose-rich diet is a risk factor for metabolic syndrome, proximal tubule injury and urolithiasis in rats,’ the authors FlisiÅ„ski et al. studied the effect of high fructose diet on kidney stone formation in male Wistar rats. The rats were fed with a regular diet (RD) or Regular diet plus drinking water with 10% fructose (F10) or 60% fructose diet (F60). The basic composition of RD and F10 diets were the same, whereas that of F60 is slightly different. The F10 group had higher fructose in the blood. Higher level of insulin, triglyceride, and insulin resistance was observed in the fructose-rich (FR) diet group and thus indicative of initiation of metabolic syndrome. Inflammation is demonstrated by showing increased excretion of MCP-1 and N-Acetyl- 24 (D)-Glucosaminidase in the FR group, lymphocytic infiltration. The gene expression of some inflammatory markers in the kidney would have supported the claim further. FR group has shown hypercalciuria, hyperphosphaturia and hypernatriuria. Kidney stones were found in two of the rats, each from the FR10 and FR60 group, whereas none were in the RD group. The article is well-written and essential in the present scenario. The authors have also included some of the limitations of the study. Although it is an initial study, the claims could have been supported by additional data. The measure of oxalate, although important but is missing in this study because of technical reasons. Here are a few suggestions
- Table1: Include average water consumption for each group.
- Although urolithiasis is more common in men than women, the authors may consider adding the use of only male rats in this study as a limitation.
- Figure 5: Not able to locate the arrow
- There was no change in uric acid levels in the blood/serum. In a clinical study, Johnson et al. 2018 observed higher uric acid after two weeks of consumption of 10% fructose water (BMC nephrology). The postprandial serum uric acid was increased after fructose intake (Cox et al. 2012 Nutr Metab (Lond). 2012;9(1):68.). Whereas in the present study, the uric acid was measured after 4 h fasting. Is this a reason why there is no change in the serum uric acid levels? Or is this species-dependent?
- Gene expression studies can further support the inflammation for inflammatory markers if the tissue samples are preserved for PCR. Alternatively, immunostaining can be done on tissue sections for inflammatory markers such as TNFa or IL6.
- The authors may consider including histological section figures for RD mice too.
Author Response
Responses to Reviewer 2:
Ad.1.The information about an average water consumption for each group has been added to Table1 in the 3rd column.
Ad.2. According to reviewer recommendation the sentence has been added in the study limitation “The present study was conducted on male rats. Due to the fact that sex hormones may influence the urine composition, the effect of FR-diet should also be confirmed in female rats.”
Ad.3. An information about the arrow on Figure 5 has been removed.
Ad.4. A sentence has been added to the discussion section “It is also thought that fructose does not increase uric acid level significantly in rats due to the presence of the uricase which causes degradation of uric acid to allantoin.”
Ad. 5. Due to the lack of histological material, we are not able to perform additional genetic or immunostaining tests suggested by the Reviewer. We agree that they would undoubtedly add to the scientific value of this work.
Ad.6. Additional figures of kidney histopathology from RD rats has been added to manuscript.
Round 2
Reviewer 2 Report
The authors have made suggested changes except for immunohistochemistry and gene expression studies due to the unavailability of samples.
Could not able to locate the words 'uricase' or 'allantoin' in reference 17 (line 277-299). The authors may recheck or change the reference.
Author Response
Dear Reviewer,
Thank you very much for your comment regarding mistakes with the reference.
I have changed the sentence by removing the word "allantoin" and give the correct reference number [10]. This sentence in changed form looks like "It is also thought that FR does not increase uric acid level significantly in rats due to the presence of the uricase which causes degradation of uric acid [10]".